# National trauma system establishment based on implementation of regional trauma centers improves outcomes of trauma care: A follow-up observational study in South Korea

**Kyoungwon Jung**[1], **Junsik Kwon**[1], **Yo Huh**[1], **Jonghwan Moon**[1], **Kyungjin Hwang**[1], **Hyun Min Cho**[2], **Jae Hun Kim**[3], **Chan Ik Park**[3], **Jung-Ho Yun**[4], **Oh Hyun Kim**[5], **Kee-Jae Lee**[6], **Sunworl Kim**[7], **Borami Lim**[7], **Yoon Kim**[8]*

**1** Department of Surgery, Ajou University School of Medicine, Suwon, South Korea, **2** Jeju Regional Trauma Center, Cheju Halla General Hospital, Jeju, South Korea, **3** Department of Trauma and Surgical Critical Care, Pusan National University College of Medicine, Busan, South Korea, **4** Department of Neurosurgery, Dankook University College of Medicine, Cheon-an, South Korea, **5** Department of Emergency Medicine, Yonsei University Wonju College of Medicine, Won-ju, South Korea, **6** Department of Information and Statistics, Korea National Open University, Seoul, South Korea, **7** National Emergency Medical Center, National Medical Center (NMC), Seoul, South Korea, **8** Department of Health Policy and Management, Seoul National University College of Medicine, Seoul, South Korea

* yoonkim@snu.ac.kr

**Data Availability Statement:** All relevant data are within the manuscript and its Supporting Information files.

## Abstract

Although South Korea is a high-income country, its trauma system is comparable to low- and middle-income countries with high preventable trauma death rates of more than 30%. Since 2012, South Korea has established a national trauma system based on the implementation of regional trauma centers and improvement of the transfer system; this study aimed to evaluate its effectiveness. We compared the national preventable trauma death rates, transfer patterns, and outcomes between 2015 and 2017. The review of preventable trauma deaths was conducted by multiple panels, and a severity-adjusted logistic regression model was created to identify factors influencing the preventable trauma death rate. We also compared the number of trauma patients transferred to emergency medical institutions and mortality in models adjusted with injury severity scores. The preventable trauma death rate decreased from 2015 to 2017 (30.5% vs. 19.9%, $p < 0.001$). In the severity-adjusted model, the preventable trauma death risk had a lower odds ratio (0.68, 95% confidence interval: 0.53–0.87, $p = 0.002$) in 2017 than in 2015. Regional trauma centers received 1.6 times more severe cases in 2017 (according to the International Classification of Diseases Injury Severity Score [ICISS]; 23.1% vs. 36.5%). In the extended ICISS model, the overall trauma mortality decreased significantly from 2.1% (1008/47 806) to 1.9% (1062/55 057) ($p = 0.041$). The establishment of the national trauma system was associated with significant improvements in the performance and outcomes of trauma care. This was mainly because of the implementation of regional trauma centers and because more severe patients were transferred to regional trauma centers. This study might be a good model for low- and middle-income countries, which lack a trauma system.

**Funding:** This study was funded by the Ministry of Health and Welfare, Republic of Korea under award number 2018-10-26 (Recipient: Yoon Kim). The funder of the study had no role in study design, data collection, data analysis, data interpretation, or writing of the report. All authors had full access to all the data in the study and had final responsibility for the decision to submit for publication.

**Competing interests:** The authors declare that they have no competing interests.

## Introduction

Injury, the leading cause of death in the reproductive-age group individuals < 40 years, is a major public health problem [1–3]. Each year, 10% of all deaths are due to injury, and many more are disabled [3, 4]. Although the magnitude of the burden of injury is alarming when compared with that of other diseases [3], the interest and investment are less than those for other public health issues [5]. The burden is especially high in low- and middle-income countries (LMICs) because more than 90% of injury-related deaths occur in LMICs [6]. Remarkable reductions in injury-related mortality, disability, and costs have been achieved in many healthcare jurisdictions by introducing trauma systems [7–12]; however, well-organized trauma systems have only been minimally implemented in most LMICs [4].

Although there are various definitions, a trauma system can be broadly defined as an organized, regional, multidisciplinary approach to trauma care [13, 14]. Moreover, the World Health Organization (WHO) and the American College of Surgeons provided consensus-based recommendations on the structure of trauma systems [4, 14–16]. Many studies have evaluated the effectiveness of trauma systems; nonetheless, it is challenging that these studies have provided low-quality evidence [17]. Although most of the abovementioned studies covered an inclusive design of trauma systems, they mainly targeted outcomes for specific cohort populations or organ injury on a state-wide level [18–20]. Recently, few examples of the establishment of national trauma systems that started in LMICs, such as African or Asian countries, have been reported; however, these trauma systems are still in their infancy, not systematic, and do not provide high-quality evidence [21–23]. Moreover, few studies have been conducted to follow up, serially and prospectively, the effect of a national trauma system (NTS) for all emergency medical institutions (EMIs) across the country.

Although South Korea is a high-income country, its trauma system is not at the same level as that of other high-income countries. Despite trauma being one of the three major causes of death in South Korea, alongside cancer and cardiovascular diseases, and one of the four major causes of death in the emergency department (ED) [2, 24], the preventable trauma death rate (PTDR) exceeded 30% by the 2010s, which was similar to that in LMICs [25–27]. To improve these circumstances, the South Korean government implemented an NTS in 2012 that was geared towards the implementation of regional trauma centers (RTCs) nationwide and improvement in the pre-hospital transfer system.

We planned a national follow-up survey every 2 years since the implementation of the NTS to assess its effectiveness. First, a survey of PTDR based on a multi-panel review was designed. Furthermore, nationwide emergency medical data were analyzed to identify changes in performance and outcomes of national trauma care.

## Materials and methods

### Emergency medical system and master plan for the establishment of a national trauma system in South Korea

The promulgation of the Emergency Medical Service Act in 1994 became the starting point for the establishment of the modern emergency medical system in South Korea; the current system was established in the 2000s [28]. The system was created by rating three levels of EMIs according to the level of available resources or specialized care. These EMIs included the Regional Emergency Medical Center (REMC), the Local Emergency Medical Center (LEMC), and the Local Emergency Medical Institution (LEMI). The REMC and LEMI represent the highest and lowest levels of EMIs, respectively. The LEMCs are also subdivided into tertiary hospitals with ≥ 500 beds and general hospitals with 300–499 as well as < 300 beds. In 2017,

there were 36 REMCs, 118 LEMCs, and 261 LEMIs, and the funding received by EMIs amounted to 250 million United States dollars (USD) [24].

The South Korean government and the health sector created a master plan for NTS establishment in 2012 (Fig 1A) [29]. The law on the establishment and operation of RTCs was enacted in the Emergency Medical Service Act and candidate institutions for RTCs, among REMCs and LEMC with $\geq$ 500 beds that met the criteria, were publicly recruited and selected after evaluation. The South Korean government provided 67 million USD per institution for the construction of facilities and equipment dedicated to the management of trauma patients and also supported labor costs for 25 dedicated trauma doctors per institution since designation. By 2017, 16 RTCs were designated, and 9 RTCs were officially opened (Fig 1B) [24, 29].

## Data used for the study

We used data from the National Emergency Department Information System (NEDIS), which collects healthcare-related information registered by EMIs nationwide in real-time based on the Emergency Medical Service Act. In 2017, 413 of 416 EMIs nationwide transmitted data. The Korean Trauma Data Bank (KTDB) was implemented in 2013 to prepare the baseline for the trauma system by collecting information transmitted from RTCs. In 2017, 14 institutions registered information.

The target population for sampling was selected from trauma deaths with at least one diagnostic code of S, T based on the South Korean Standard Classification of Diseases (the South Korean version of the International Classification of Disease) of 2015 and 2017 according to the NEDIS (Fig 2A). All medical records, including imaging studies, were obtained from the EMIs with cooperation from the central and local governments. Pre-hospital information, including the data of hospitals that transferred patients were also included.

## National survey of the preventable trauma death rate by multi-panel review

From 2015, we performed a national survey on PTDR every 2 years. The investigation was conducted in the following order: design/extraction of the sample population, data collection, panel review, reliability test for the review process, and result analysis. The criteria and review process for the decision on preventability were mainly based on the WHO "Guidelines for trauma quality improvement programmes (TQIP)" [4].

The "Guidelines for TQIP" by the WHO provides how-to-do guidance on a range of different trauma quality improvement methods. These are broadly applicable to all health care institutions that care for the injured in countries at all economic levels. One or more of the methods described in this document will be directly applicable to any given institution and will enable that institution to upgrade the level of function of its existing trauma quality improvement activities. This would help strengthen the quality of trauma care and save the lives of many injured persons.

The criteria for preventing trauma deaths were based on WHO guidelines for TQIP [4]. The main factors underlying the decision regarding the preventability of trauma deaths comprised severity of injuries and appropriateness of trauma care. Definitions were as follows.

- "Preventable trauma deaths" were deaths that could have been prevented if appropriate steps had been taken, with accompanying injuries and sequelae considered survivable. These cases had frank deviations from the standard of care that, directly or indirectly, caused the patient's death.

- "Potentially preventable trauma deaths" were deaths that potentially could have been prevented if appropriate steps had been taken, with accompanying injuries and sequelae

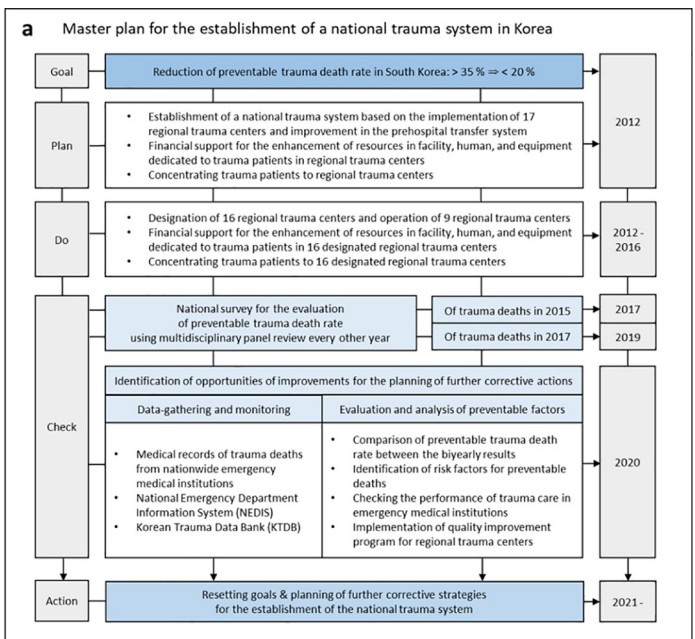

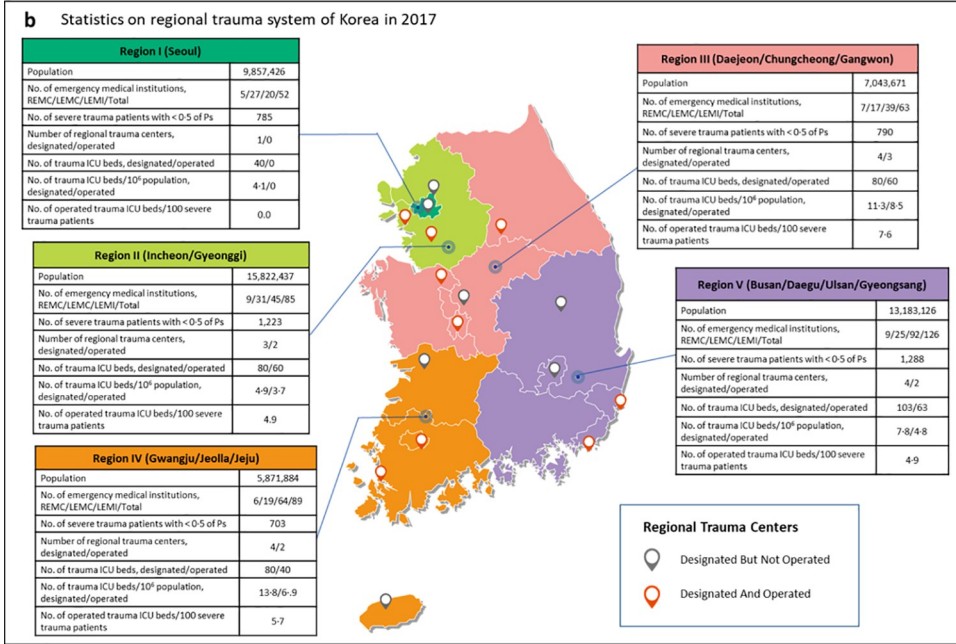

**Fig 1. Master plan for the establishment of a national trauma system in South Korea.** (a) shows the master plan for the establishment of a national trauma system in South Korea. The South Korean government and medical society created the master plan in 2012. It was based on the implementation of regional trauma centers and improvement in the prehospital transfer system for transferring the majority of severe trauma patients to trauma centers. (b) shows that 16 regional trauma centers were designated, and nine regional trauma centers were officially operational by 2017 after a preparation period of 1–3 years from 2012 to 2016. It also shows the number of trauma intensive care unit beds per population and severe trauma patients in 2017 according to the five regions (divided in consideration of the population and the administrative district classification in South Korea). No, number; REMC, Regional Emergency Medical Center; LEMC, Local Emergency Medical Center; LEMI, Local Emergency Medical Institution; ICU, intensive care unit.

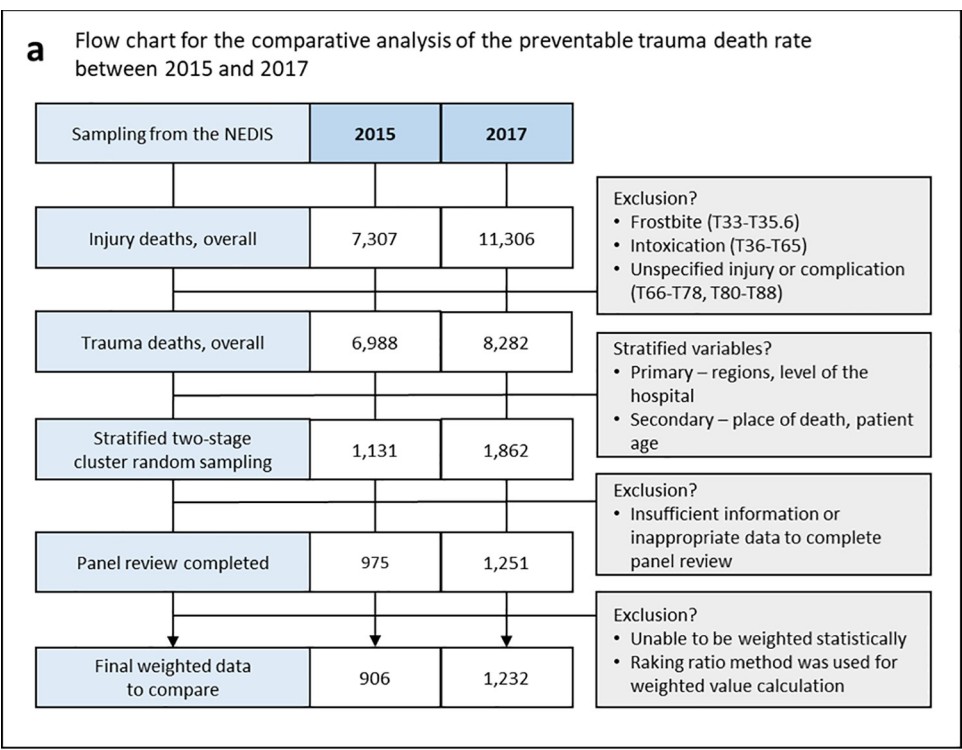

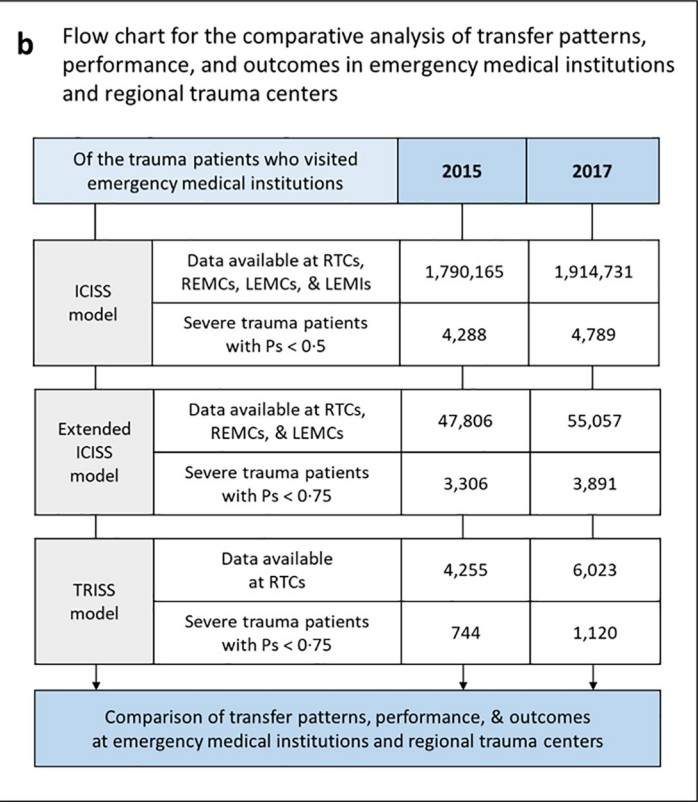

**Fig 2. A flow chart of the study design.** (a) is a flow chart for the comparative analysis of the preventable trauma death rate via a multi-panel review; (b) is a flow chart for the analysis of the changes in performance, including the transfer patterns of trauma patients and the outcomes of trauma care in the emergency medical system in South Korea. RTC, Regional Trauma Center; REMC, Regional Emergency Medical Center; LEMC, Local Emergency Medical

Center; LEMI, Local Emergency Medical Institution; ICU, intensive care unit; NEDIS, National Emergency Department Information System; ICISS, International Classification of Disease Injury Severity Score; Ps, probability of survival; TRISS Trauma and Injury Severity Score.

considered severe but survivable. These cases had some deviations from the standard of care that, directly or indirectly, caused the patient's death.

- "Non-preventable trauma deaths" were unavoidable deaths, as accompanying injuries and sequelae were considered non-survivable even with optimal management. Evaluation and management were appropriate according to accepted standards. If the patient had co-morbid factors that were major contributors to death, such cases were considered non-preventable trauma deaths.

We used a structured review form made according to the WHO TQIP guidelines, including audit filters, for the review of medical records (S1 Appendix). Designated assistants (pre-reviewers), comprising trauma coordinators working in RTCs nationwide, investigated and recorded the general characteristics of patients, injury-related information, as well as transport and treatment-related information, using the review form before the preventable death panel review. The case review panels mainly comprised trauma specialists working at RTCs. A total of 10 teams in 2015 and five teams in 2017 were formed. Moreover, a committee with five trauma specialties was responsible for developing guidelines for the entire review process and training reviewers. The committee reviewed and confirmed the final decisions when the multi-panel review could not achieve consensus. Three teams were selected to evaluate the reliability of the panel review. They repeated reviews for 5% of the overall cases that had already been reviewed by other teams (S2 Appendix).

## National evaluation of the performance and outcomes of the new trauma system

An analysis was conducted to compare the performances and outcomes of trauma care between two survey periods. International Classification of Disease Injury Severity Score (ICISS), extended ICISS, and Trauma and Injury Severity Score (TRISS) models were created for severity adjustments.

The ICISS took an empirical estimation approach to injury severity scoring with the International Classification of Disease (ICD) survival probability formulation. Probability of survival (Ps) is an ICD code-specific estimate of the survival probability associated with that particular injury. The traditional ICISS is calculated as the product of Ps for as many as 10 injuries and ranges from 0 (unsurvivable) to 1 (null probability of death). Other versions of the ICISS include Ps of the worst injury and independent Ps calculated on patients with isolated injuries [30, 31].

The TRISS methodology, which was introduced in the 1980s, is the most widely used formula for calculating the Ps of an individual trauma patient. It calculates the probability of survival of a trauma patient by using a formula that includes the mechanism of injury (blunt or penetrating), age (older or younger than 55 years), Injury Severity Score (ISS), and Revised Trauma Score (RTS). Once the probability of survival of the individual patient is calculated, the expected survival and number of deaths for a group of patients can be determined [30, 32].

We calculated the Ps from these models for the outcome analysis (Fig 2B). The extended ICISS indicated an ICISS model adjusted for age and the RTS. To calculate the RTS, the initial physiologic parameters on ED admission were used. The LEMIs were excluded from the extended ICISS model because physiologic parameters in LEMIs were not registered to

NEDIS. Since the injury severity scores required for the TRISS model are only registered in the KTDB, it was possible to create TRISS only with data collected from RTCs.

## Statistical analysis

To calculate the nation's representative PTDR and improve the efficiency of the panel review, we selected the target of the trauma death review through stratified two-stage cluster sampling (S3 Appendix). The stratification was designed as a double layer; the primary stratified variables were region and EMI level, and the secondary stratified variables were place (timing) of death and patient age. Our initial targeted sample sizes were 1,000 in 2015 and 1,300 in 2017; however, considering the cases to be excluded from the panel review, the survey sample sizes were determined to be 1,131 and 1,862, respectively. The sample size was targeted such that stable estimation would be possible to meet a limit of error of approximately ±4.5%p (2015) and ±3.8%p (2017) at 95% confidence levels for population ratio estimation.

To estimate the population PTDR, the sample weights of each hospital level and death were calculated according to the sample design method and applied to analyze the sample-designed survey data. For continuous data, normality testing was performed using the Kolmogorov-Smirnov test. Categorical variables were compared using the Chi-Square and Fisher's exact tests accordingly. Comparison between groups was performed using the t-test and analysis of variance. The paired t-test was used to compare differences in 2015 and 2017. To examine the association between the likelihood of preventable death and its relating factors, we constructed multivariate logistic regression models. The goodness of fit for the models was confirmed through the Hosmer-Lemeshow test. The agreement between panel teams was evaluated using Cohen's Kappa index. All statistical analyses were performed using SPSS (Version 18.0; IBM Corp., Armonk, NY), and two-sided $p$ values $< 0.05$ were considered to indicate statistical significance.

## Ethics approval and consent to participate

This study was approved by the Institutional Review Board (IRB) of Seoul National University (IRB No. E-1811-005-982). Informed consent was waived by the board due to the observational nature of the study.

## Results

### Change in the preventable trauma death rate and influencing factors

Overall, trauma deaths comprised 6,988 deaths from 355 EMIs in 2015 and 8,282 deaths from 368 EMIs in 2017 registered in the NEDIS. Of these, the number of sample cases included for the PTDR review was 975 (14.0%) from 60 EMIs in 2015 and 1,251 (15.1%) from 117 EMIs in 2017; the number of final weighted data for comparative analysis was 906 in 2015 and 1,232 in 2017. In the reliability test for the panel review, the Kappa index was 0.49 in 2015 and 0.61 in 2017, indicating moderate and substantial agreements, respectively.

The PTDR in 2017 was lower than that in 2015 (19.9% vs. 30.5%, $p$ <0.001) (Table 1, Fig 3A). The PTDR in the group after admission to the first hospital decreased significantly from 2015 to 2017 (37.1% vs. 18.2%, $p$ <0.001). In terms of regions, the PTDR significantly decreased in all regions except Region I, where there was a minimal change from 30.8% in 2015 to 30.2% in 2017 (Table 1, Fig 3B). Regarding EMIs that were the destinations of hospitalization, the LEMCs and LEMIs presented significantly lower PTDRs in 2017 (all $p$ <0.001). The PTDR significantly decreased in the no transfer group from 2015 to 2017 (28.4% vs. 15.5%, $p$ <0.001).

**Table 1. Comparison of basic characteristics of the study patients and the preventable trauma death rate\*.**

| | Study Trauma Patients (%) | | | Preventable Trauma Deaths (%) | | |
|---|---|---|---|---|---|---|
| | 2015 | 2017 | *p* value | 2015 | 2017 | *p* value |
| **Total** | **906 (100.0)** | **1,232 (100.0)** | | **276 (30.5)** | **245 (19.9)** | **<0.001** |
| Sex | | | 0.570 | | | |
| Male | 628 (69.3) | 868 (70.5) | | 186 (29.6) | 172 (19.8) | <0.001 |
| Female | 278 (30.7) | 364 (29.5) | | 91 (32.6) | 73 (20.1) | <0.001 |
| Age, years† | | | 0.649 | | | |
| ≤14 | 16 (1.8) | 16 (1.3) | | 2 (12.5) | 3 (17.6) | 1.000 |
| 15–54 | 263 (29.0) | 352 (28.6) | | 66 (25.2) | 56 (15.9) | 0.004 |
| ≥55 | 627 (69.2) | 864 (70.1) | | 208 (33.2) | 186 (21.5) | <0.001 |
| Place of death | | | 0.049 | | | |
| DOA at the first hospital | 164 (18.1) | 284 (23.1) | | 7 (4.3) | 4 (1.4) | 0.060 |
| At the ED of the first hospital | 101 (11.1) | 123 (10.0) | | 33 (33.0) | 46 (37.4) | 0.495 |
| After hospitalization at the first hospital | 394 (43.5) | 506 (41.1) | | 146 (37.1) | 92 (18.2) | <0.001 |
| Transfer to the second or additional hospitals | 247 (27.3) | 319 (25.9) | | 90 (36.3) | 102 (32.0) | 0.281 |
| Region | | | 0.762 | | | |
| I (Seoul) | 124 (13.7) | 189 (15.3) | | 38 (30.8) | 57 (30.2) | 0.927 |
| II (Incheon/Gyeonggi) | 206 (22.7) | 275 (22.3) | | 56 (27.4) | 46 (16.7) | 0.005 |
| III (Daejeon/Chungcheong/Gangwon) | 181 (20.0) | 227 (18.4) | | 47 (26.0) | 34 (15.0) | 0.006 |
| IV (Gwangju/Jeolla/Jeju) | 163 (18.0) | 216 (17.5) | | 66 (40.7) | 56 (25.9) | 0.002 |
| V (Busan/Daegu/Ulsan/Gyeongsang) | 232 (25.6) | 325 (26.4) | | 68 (29.4) | 52 (16.0) | <0.001 |
| Type of emergency institution | | | < 0.001 | | | |
| RTC | 222 (24.5) | 381 (30.9) | | 46 (20.7) | 73 (19.2) | 0.442 |
| Designated and operated | 73 (8.1) | 265 (21.5) | | 16 (21.9) | 42 (15.9) | 0.229 |
| Designated, but not operated | 150 (16.6) | 117 (9.5) | | 31 (20.7) | 31 (26.5) | 0.263 |
| REMC | 75 (8.3) | 232 (18.8) | | 25 (33.8) | 55 (23.7) | 0.086 |
| LEMC | 482 (53.2) | 483 (39.2) | | 166 (34.4) | 106 (22.0) | <0.001 |
| Tertiary hospital with ≥ 500 beds | 304 (33.6) | 116 (9.4) | | 111 (36.5) | 27 (23.3) | 0.010 |
| General hospital with 300–499 beds | 120 (13.2) | 302 (24.5) | | 34 (28.1) | 65 (21.5) | 0.150 |
| General hospital with < 300 beds | 57 (6.3) | 64 (5.2) | | 21 (36.8) | 14 (21.0) | 0.070 |
| LEMI | 127 (14.0) | 136 (11.0) | | 39 (30.5) | 11 (8.0) | <0.001 |
| Type of injury | | | 0.716 | | | |
| Blunt | 880 (97.1) | 1,190 (96.7) | | 269 (30.6) | 239 (20.1) | <0.001 |
| Penetrating | 6 (0.7) | 12 (1.0) | | 1 (14.3) | 2 (16.7) | 1.000 |
| Other/unknown | 20 (2.2) | 29 (2.4) | | 6 (30.8) | 3 (10.3) | 0.133 |
| Transfer from another hospital‡ | | | 0.290 | | | |
| No | 612 (67.9) | 902 (73.2) | | 174 (28.4) | 140 (15.5) | <0.001 |
| Yes | 249 (32.1) | 330 (26.8) | | 85 (34.1) | 105 (31.8) | 0.556 |

\*All data are shown as numbers (percentages). All percentage values are weighted.

† Mean±standard deviation for age was 62.1±20.7 years in 2015 and 62.9±19.6 years in 2017.

‡ There were 45 cases with an unknown transfer status in 2015.

DOA, dead on arrival; RTC, Regional Trauma Center; REMC, Regional Emergency Medical Center; LEMC, Local Emergency Medical Center; LEMI, Local Emergency Medical Institution.

In the most valid logistic regression model, ICISSs, RTSs, and Region IV were found to be factors that significantly influenced preventable death (Fig 4). In the severity-adjusted model, the risk of preventable death indicated a significantly lower odds ratio (0.68, 0.53–0.87) in

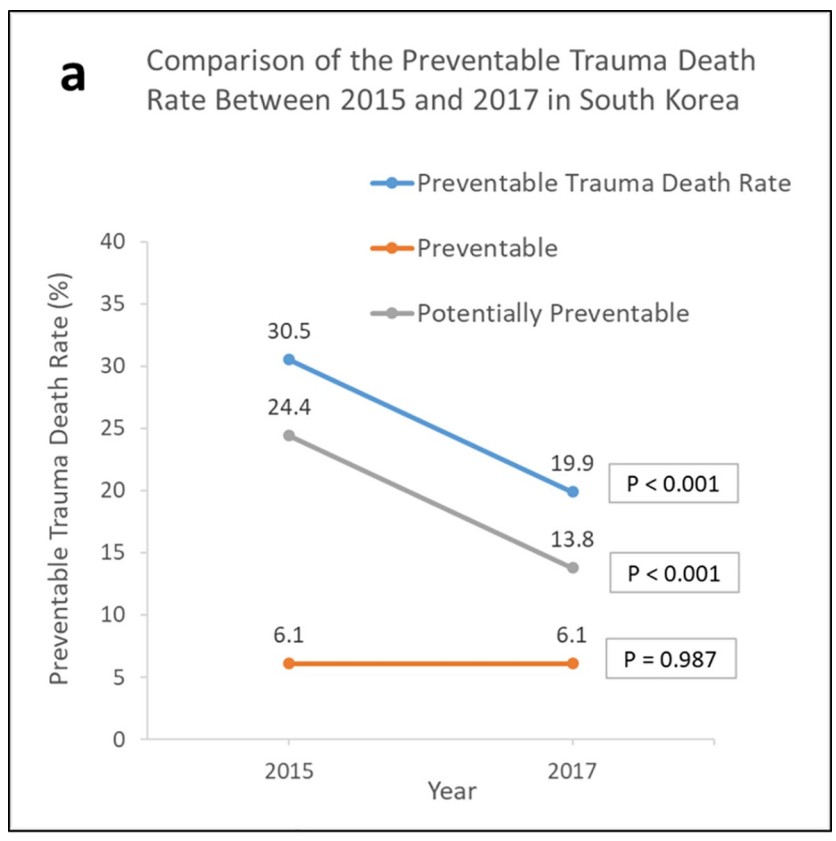

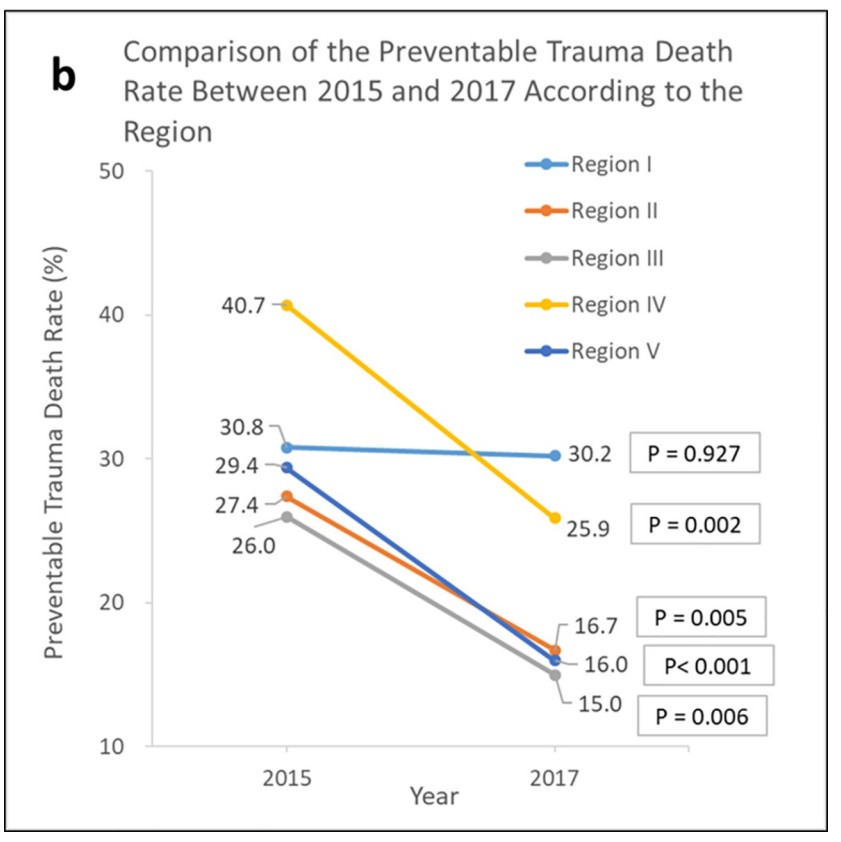

**Fig 3. Comparison of the preventable trauma death rates between 2015 and 2017.** (a) and (b) show the results of the comparison of the preventable trauma death rates according to a multi-panel review (between 2015 and 2017).

2017 than in 2015. Additionally, Region III had a significantly different odds ratio (0.58, 0.37–0.89) than Region I.

## Changes in the transfer patterns and outcomes of trauma patients

According to the NEDIS, 1,790,165 and 1,914,731 patients with trauma in 2015 and 2017, respectively, visited EMIs. Of these, 4,288 (0.24%) in 2015 and 4,789 (0.25%) in 2017 were severe trauma patients who had Ps < 0.5 according to the ICISS model (Table 2). Based on the extended ICISS model, there were 3,306 (6.9%) severe cases in 2015 and 3,891 (7.1%) in 2017 with Ps < 0.75 among trauma patients (47,806 in 2015 and 55,057 in 2017) who visited RTCs, REMCs, and LEMCs. In RTCs, 744 (17.5%) severe trauma patients with Ps < 0.75 in 2015 and 1,120 (18.6%) in 2017 were treated based on the TRISS model. Distinguishing per the type of EMI, 11.3% of all trauma patients in 2015 visited an REMC or RTC; the percentage increased to 19.7% in 2017 (Fig 5A). Meanwhile, the number of ICISS-based severe trauma patients increased by 21.6% points (from 30.7% in 2015 to 52.3% in 2017) in REMCs and RTCs. In particular, the percentage of severe trauma patients received in RTCs in 2017 (36.5%) was 1.6 times higher than that received in 2015 (23.1%); thus, accounting for more than 1/3 of the total number of severe cases (Fig 5B).

Compared with that in 2015, the severity-adjusted trauma mortality rate was lower in 2017. In the extended ICISS model, the overall mortality decreased significantly from 2.1% (1,008/

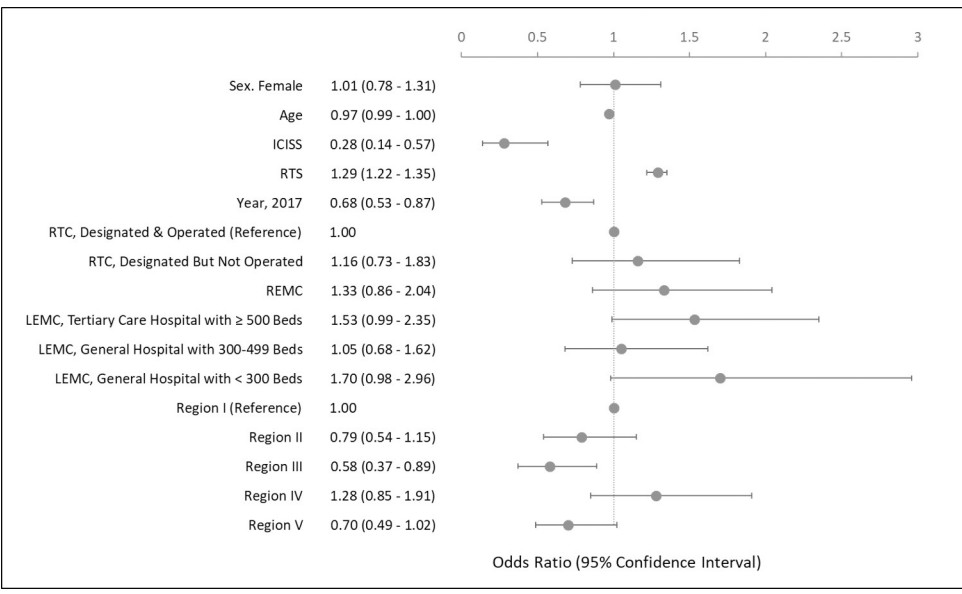

**Fig 4. Multivariate analysis of risk factors for preventable trauma deaths.** Baseline characteristics were compared between two groups using the chi-square or Fisher's exact tests. Logistic regression was used to explain the relationship between preventable trauma deaths and influencing factors. We confirmed that the goodness of fit for logistic regression was satisfactory using the Hosmer–Lemeshow test. Two-sided *P* values of 0.05 or less were considered to indicate statistical significance. The likelihood of preventable trauma death according to related factors was expressed as odds ratios and 95% confidence intervals relative to the reference value. ICISS, International Classification of Disease Injury Severity Score; RTS, Revised Trauma Score; RTC, Regional Trauma Center; REMC, Regional Emergency Medical Center; LEMC, Local Emergency Medical Center.

**Table 2. Comparison of the number of trauma patients according to the type of emergency medical institution\*.**

| | No. of EMIs | | ICISS Model | | | | Extended ICISS Model | | | | | |
| --- | --- | --- | --- | --- | --- | --- | --- | --- | --- | --- | --- | --- |
| | | | No. of Total Trauma Patients | | No. of Severe Trauma Patients (Ps of <0.5) | | No. of Total Trauma Patients | | No. of Severe Trauma Patients (Ps of <0.25) | | No. of Severe Trauma Patients (Ps of 0.25–0.75) | |
| | 2015 | 2017 | 2015 | 2017 | 2015 | 2017 | 2015 | 2017 | 2015 | 2017 | 2015 | 2017 |
| **RTC** | **13 (3.1)** | **16 (3.8)** | **126,401 (7.1)** | **155,015 (8.1)** | **989 (23.1)** | **1,749 (36.5)** | **9,000 (18.8)** | **13,053 (23.7)** | **406 (30.1)** | **857 (57.1)** | **529 (27.0)** | **932 (39.0)** |
| Designated/operated | 4 (1.0) | 9 (2.2) | 46,697 (2.6) | 94,415 (4.9) | 272 (6.3) | 1,238 (25.9) | 3,399 (7.1) | 8,680 (15.8) | 173 (12.8) | 436 (29.1) | 161 (8.2) | 643 (26.9) |
| Designated/not operated | 9 (2.1) | 7 (1.7) | 79,704 (4.5) | 60,600 (3.2) | 717 (16.7) | 511 (10.7) | 5,601 (11.7) | 4,823 (8.8) | 233 (17.3) | 180 (12.0) | 368 (18.8) | 289 (12.1) |
| **REMC** | **9 (2.1)** | **23 (5.5)** | **75,940 (4.2)** | **223,058 (11.6)** | **326 (7.6)** | **757 (15.8)** | **4,210 (8.8)** | **11,225 (20.4)** | **102 (7.6)** | **241 (16.1)** | **168 (8.6)** | **479 (20.0)** |
| **LEMC** | **125 (29.7)** | **116 (27.9)** | **868,727 (48.5)** | **774,671 (40.5)** | **2,359 (55.0)** | **1,740 (36.3)** | **34,596 (72.4)** | **30,329 (55.1)** | **841 (62.3)** | **643 (42.9)** | **1,264 (64.5)** | **980 (41.0)** |
| Tertiary hospital with ≥500 beds | 30 (7.1) | 16 (3.8) | 261,938 (14.6) | 124,332 (6.5) | 930 (21.7) | 389 (8.1) | 11,995 (25.1) | 5,604 (10.2) | 307 (22.8) | 152 (10.1) | 537 (22.8) | 241 (10.1) |
| General hospital with 300–499 Beds | 68 (16.2) | 69 (16.6) | 477,758 (26.7) | 481,243 (25.1) | 1,184 (27.6) | 1,068 (22.3) | 18,396 (38.5) | 18,853 (34.2) | 422 (31.3) | 367 (24.5) | 615 (31.3) | 590 (24.5) |
| General hospital with <300 Beds | 27 (6.4) | 31 (7.5) | 129,031 (7.2) | 169,096 (8.8) | 245 (5.7) | 283 (5.9) | 4,205 (8.8) | 5,872 (10.7) | 112 (8.3) | 108 (8.3) | 112 (8.3) | 149 (8.3) |
| **LEMI** | **274 (65.1)** | **261 (62.7)** | **917,097 (40.2)** | **761,987 (39.8)** | **614 (14.3)** | **541 (11.3)** | n/a | n/a | n/a | n/a | n/a | n/a |
| **Total** | **421 (100.0)** | **416 (100.0)** | **1,790,165 (100.0)** | **1,914,731 (100.0)** | **4,288 (100.0)** | **4,789 (100.0)** | **47,806 (100.0)** | **55,057 (100.0)** | **1,349 (100.0)** | **1,500 (100.0)** | **1,957 (100.0)** | **2,391 (100.0)** |

\* All data are shown as numbers (percentages).

ICISS, International Classification of Diseases Injury Severity Score; TRISS, Trauma and Injury Severity Score; EMI, emergency medical institution; Ps, probability of survival; RTC, Regional Trauma Center; REMC, Regional Emergency Medical Center; LEMC, Local Emergency Medical Center; LEMI, Local Emergency Medical Institution; n/a, not available.

47,806) in 2015 to 1.9% (1,062/55,057) in 2017 ($p = 0.041$). The reduction rate was the greatest in severe cases with Ps < 0.25 from 2015 to 2017, although insignificant (53.2% vs. 49.9%, $p = 0.079$) (Fig 5C). In the TRISS model, the overall mortality rate of severe trauma patients did not significantly change from 2015 to 2017 (4.7% to 4.4%; $p = 0.455$), and the mortality rate of cases with Ps < 0.25 decreased by 6.3%, although insignificant (56.5% vs. 50.2%, $p = 0.091$) (Fig 5D).

## Discussion

This study aimed to evaluate the effects of the attempts to implement and establish a national trauma system in South Korea, which shows a high preventable trauma mortality rate (>30%), comparable to that of LMICs. We planned to serially conduct national surveys every two years for the purpose of reducing the preventable trauma death rate, which was the original goal when the national trauma system project was first developed at the end of 2012. Moreover, the transfer status and outcome changes of trauma patients were investigated using the NEDIS, which is registered by over 400 emergency medical institutions nationwide. In this study, severe trauma patients were more concentrated in the regional trauma centers, and the preventable trauma death rate decreased by more than 10% in only 2 years after the government-led trauma system establishment. Furthermore, the overall performance and outcomes of trauma care improved in all the emergency medical institutions nationwide.

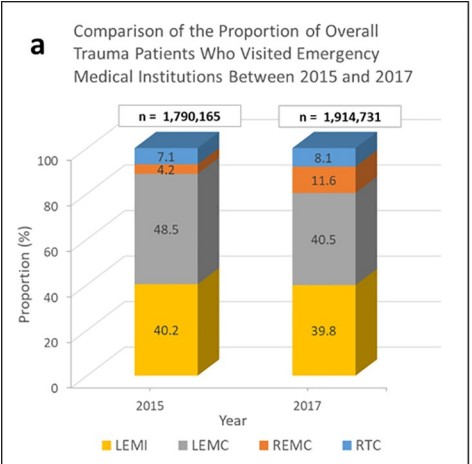

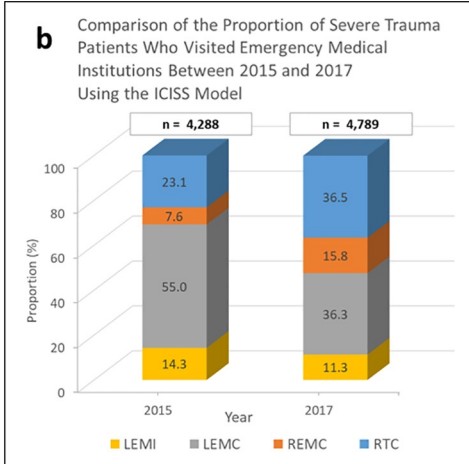

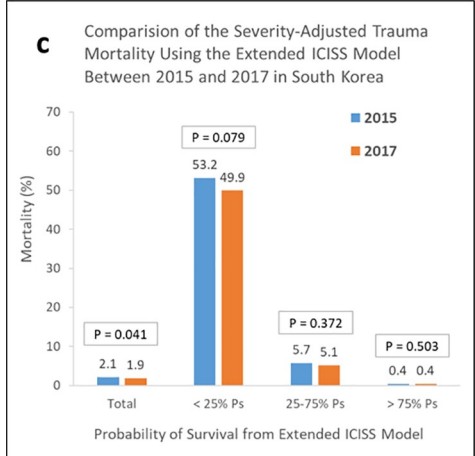

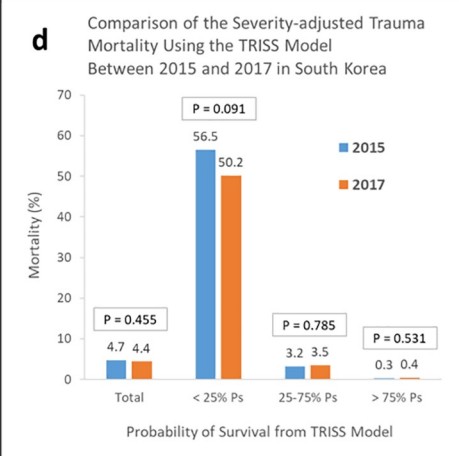

**Fig 5. Changes in performance and outcomes of trauma patients between 2015 and 2017.** (a) and (b) are based on the ICISS model targeted to all emergency medical institutions in South Korea. (c) shows the results of the severity-adjusted trauma mortality model using an extended ICISS model targeted for emergency medical institutions other than LEMIs. (d) is a TRISS model resulting from data registered by RTCs. RTC, Regional Trauma Center; REMC, Regional Emergency Medical Center; LEMC, Local Emergency Medical Center; LEMI, Local Emergency Medical Institution; ICISS, International Classification of Disease Injury Severity Score; TRISS, Trauma and Injury Severity Score; Ps, probability of survival.

Although it is emphasized that the trauma system should be built as an inclusive design [7–12], it is not easy to do so on a national level. Furthermore, it is difficult to prove the effectiveness of trauma systems. Studies have evaluated the effectiveness of the NTS; however, most studies have provided low-quality evidence [17]. The biggest problem has been adequately adjusting for referral bias. In this study, we adopted a different approach; the PTDR was estimated through multi-panel reviews since the purpose of this project was to lower the PTDR in South Korea; analysis was performed on all trauma patients using the existing database.

Preventability judgments may be subjective, and there may be variations in reliability between different panel assessments; nonetheless, death panel reviews remain a straightforward method of accomplishing the goal of assessing and improving the quality of care [4]. Despite the lack of quantitative precision, these reviews are often a major stimulus for improvements in trauma care [33, 34]. In this study, the definition of preventability and the review methodology essentially followed the WHO TQIP guidelines, and the results of

reliability evaluations among team panels showing moderate to the substantial agreement were comparable to those of previous studies [25–27, 35].

Here, we noted that the PTDR significantly decreased by 10% from 2015 to 2017, indicating that the national goal of decrease to < 20% by 2020 has already been achieved. However, we realized that many factors could be improved because the rate of definitively preventable deaths did not decrease compared with that of potentially preventable deaths. One of the prominent points in the comparison of the PTDR was the differences among regions. There was no PTDR difference in Region I (Seoul), whereas the remaining four regions experienced a significant decrease, showing similar declines of 10%–15%. This indicates that the trauma care system has not improved in Seoul, the only region where RTCs have not hitherto operated. In severity-adjusted models, Region III had a significantly lower odds ratio than Region I for PTDR, probably because Region III employed the greatest number of beds according to the population ($8.5$ beds/$10^6$) and the number of trauma ICU beds per the number of patients with severe trauma (7.6 beds/100) in 2017.

The role of RTCs in decreasing the PTDR between 2015 and 2017 was not statistically remarkable. This may be owing to the limitations of the PTDR calculation method. A type of EMI, which was recorded as a final-destined hospital for trauma deaths, was not attributed solely to the responsibility of PTDR. Rather, we had to consider the injury severity and errors occurring at the pre-hospital or inter-hospital stage. In the severity-adjusted model, the RTCs tended to have lower odds ratio than other EMIs, although insignificant. Nevertheless, the effectiveness of the NTS establishment in South Korea is evident as the year 2017 had a lower odds ratio of PTDR at 68% than 2015.

The national trauma incidence increased by 125,000 cases in 2 years, and the number of severe cases with Ps < 50% increased by 500. These trauma patients were commonly transferred to RTCs; this trend was more pronounced for severe cases (at least 50% were transferred to REMCs or RTCs and at least 33% to RTCs). The outcome of severity-adjusted trauma mortality was better in 2017 than in 2015. Severe trauma patients with lower Ps showed greater outcome improvements in EMIs other than LEMIs; however, the number of cases was insufficient to show a significant reduction in RTCs. Consequently, it is assumed that the transfer of severe cases to RTCs increased due to trauma system establishment; this would have contributed to outcome improvements.

There has been controversy regarding the appropriate number of trauma centers, even in the United States of America (USA), where an inclusive trauma system has been established for more than 50 years [36–40]. Although a different standard should be applied in South Korea, which has different injury mechanisms and a different geographical environment from the USA, 17 RTCs seem insufficient to cover trauma patients nationwide. Nevertheless, we believe that the implementation of these RTCs is the starting point for trauma outcome improvement. Establishing these RTCs would be the basis for building an inclusive NTS when integrating with other EMIs. When enacting the law to establish trauma systems, the South Korean government announced its plan to extend the number and scale of RTCs and included the designation of lower-level trauma centers that would be considered for recruitment among REMC and LEMC with > 300 beds. If the NTS is further established in this way, approximately 90% of severe trauma patients will be covered by the system.

This study has some limitations. First, the retrospective study design precluded the analysis of unrecorded factors or missing values. Especially it was impossible to use negative controls to back up the causal interpretation of the study findings because we could not adjust injury severity appropriately due to the limited quality of information and data collected retrospectively for trauma death cases. Considering these points, we chose a method of judging directly through panel reviews, and quantitative analysis using big data analysis was added to

compensate for the limitations of subjectivity resulting from this. Second, the evaluation of preventability relied entirely on analysis by expert panels, which has limitations in objective reproducibility. Third, there was a time lag between master plan development in 2012 and the actual implementation of the national trauma system, and we only focused on two-time points (in 2015 and 2017). We believe it will take a 2-year interval to allow the emergency medical system and the emergency medical institutions to adjust to its new policies and implement new systems. Therefore, we planned a national follow-up survey every 2 years since the implementation of the national trauma system to confirm its effectiveness and consequently. Fourth, there were slight differences in sampling methods and panel reviews between 2015 and 2017. The number of EMIs and cases targeted for sampling increased in 2017. The panels had to visit EMIs to review medical records, and follow-up review was possible only for data recorded on the review forms in 2015, whereas all data from EMIs were reviewed in one place in 2017. Thus, repeated reviews were possible for all data in 2017. Hence, 10 teams were required for panel review in 2015 and only five teams in 2017. Fifth, the extended ICISS model could not be applied for LEMIs because LEMIs did not register initial ED physiologic parameters in the NEDIS. Sixth, the standards of Ps for severe cases in the severity-adjusted models ($<0.5$ in the ICISS, $<0.25$ in the extended ICISS and TRISS) were arbitrary. They were decided by the authors considering that higher standards should be applied to higher-level EMIs. Finally, we replaced age group, transfer (yes/no), and place of death with continuous values of age, RTS, and ICISS when creating the logistic regression model. This was owing to the difficulty in obtaining a statistically valid model when the stratification variables used for sampling were applied to the logistic regression model.

Despite these limitations, evidence from this study and previous research indicates that trauma outcomes can improve by establishing trauma centers. In this study, severe trauma patients were more concentrated in the regional trauma centers, and the preventable trauma death rate decreased by >10% in only 2 years after the establishment of the government-led trauma system, indicating that through national efforts, trauma performance and outcomes can improve within a short timeframe after the trauma system is implemented.

Furthermore, trauma care's overall performance and outcomes improved in all emergency medical institutions nationwide. These findings provide evidence of the effectiveness of establishing a national trauma system and a protocol for assessing this effectiveness, especially in LMICs with a large burden of injury that do not hitherto have an effective trauma system.

## Conclusions

The government-led establishment of the NTS (including financial investment) was associated with a significant improvement in the performance and outcomes of trauma care nationwide in South Korea. Trauma outcomes improved as RTCs became operational nationwide, and more severe cases were transferred to RTCs. Based on these results, we can expect a further decrease in the PTDR in South Korea when RTCs mature, and more severe cases are concentrated at EMIs with high levels of trauma care. The results of this study may provide a good model for LMICs currently lacking a trauma system.

## Supporting information

**S1 Appendix. The structured review form, including audit filters for the multi-panel review.**
(PDF)

**S2 Appendix. Process of multi-panel review for preventable trauma death rate.**
(PDF)

**S3 Appendix. Summary of sampling method and estimation of traumatic deaths.**
(PDF)

**S4 Appendix. Data for comparison of basic characteristics of the study patients and the preventable trauma death rate.**
(XLSX)

## Author Contributions

**Conceptualization:** Kyoungwon Jung, Junsik Kwon, Yo Huh, Jonghwan Moon, Hyun Min Cho, Jae Hun Kim, Chan Ik Park, Jung-Ho Yun, Oh Hyun Kim, Yoon Kim.

**Data curation:** Kyoungwon Jung, Junsik Kwon, Yo Huh, Jonghwan Moon, Kyungjin Hwang, Hyun Min Cho, Jae Hun Kim, Chan Ik Park, Jung-Ho Yun, Oh Hyun Kim, Kee-Jae Lee, Sunworl Kim, Borami Lim, Yoon Kim.

**Formal analysis:** Kyoungwon Jung, Junsik Kwon, Yo Huh, Jonghwan Moon, Kyungjin Hwang, Hyun Min Cho, Jae Hun Kim, Chan Ik Park, Jung-Ho Yun, Oh Hyun Kim, Kee-Jae Lee, Sunworl Kim, Borami Lim, Yoon Kim.

**Funding acquisition:** Yoon Kim.

**Investigation:** Kyoungwon Jung, Junsik Kwon, Yo Huh, Jonghwan Moon, Kyungjin Hwang, Hyun Min Cho, Jae Hun Kim, Chan Ik Park, Jung-Ho Yun, Oh Hyun Kim, Yoon Kim.

**Methodology:** Kyoungwon Jung, Kee-Jae Lee, Sunworl Kim, Borami Lim, Yoon Kim.

**Project administration:** Borami Lim, Yoon Kim.

**Supervision:** Kyoungwon Jung, Yoon Kim.

**Validation:** Kyoungwon Jung, Junsik Kwon, Yoon Kim.

**Visualization:** Kyoungwon Jung.

**Writing – original draft:** Kyoungwon Jung, Sunworl Kim, Yoon Kim.

**Writing – review & editing:** Kyoungwon Jung, Junsik Kwon, Yo Huh, Jonghwan Moon, Kyungjin Hwang, Hyun Min Cho, Jae Hun Kim, Chan Ik Park, Jung-Ho Yun, Oh Hyun Kim, Kee-Jae Lee, Sunworl Kim, Yoon Kim.

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
