## [Editor Report · Decision Letter 0]

7 Jul 2021

 PGPH-D-21-00240 National trauma system establishment based on implementation of regional trauma centers improves outcomes of trauma care; A follow-up observational study in South Korea PLOS Global Public Health

Dear Dr. Kim,

Thank you for submitting your manuscript to PLOS Global Public Health. After careful consideration, we feel that it has merit but does not fully meet PLOS Global Public Health’s publication criteria as it currently stands. Therefore, we invite you to submit a revised version of the manuscript that addresses the points raised during the review process.

We look forward to receiving your revised manuscript.

Kind regards,

Jagdish Khubchandani, MBBS, PhD

Academic Editor

Additional Editor Comments (if provided):

Thank you for your submission to PLOS GLOBAL HEALTH. A few minor concerns are as follows-

1. The data presented is from 2015 to 2017, but the system was established in 2012. Why not present the data from 2012 to 2020?

2. Were any weighted analyses conducted? Why not? How was power analyses conducted for sample size? based on what effects?

3. The manuscript at times runs like a long thesis and such redundancy should be cut down.

4. When you mention Korea- replace it with South Korea throughout the paper.

5. What are the implications for practice? and future research?

6, There are many other limitations concerning reliability and validity - explain these and how would the limitations affect the results? What else could be done to improve the rigor of the study?
---

## [Decision Letter · Decision Letter 1]

29 Oct 2021

PGPH-D-21-00240R1

National trauma system establishment based on implementation of regional trauma centers improves outcomes of trauma care; A follow-up observational study in South Korea

Dear Dr. Kim,

Thank you for submitting your manuscript to PLOS Global Public Health. After careful consideration, we feel that it has merit but does not fully meet PLOS Global Public Health’s publication criteria as it currently stands. Therefore, we invite you to submit a revised version of the manuscript that addresses the points raised during the review process.

We look forward to receiving your revised manuscript.

Kind regards,

Jagdish Khubchandani

Academic Editor

Journal Requirements:

Additional Editor Comments (if provided):

Reviewers' comments:

Reviewer's Responses to Questions

**Comments to the Author**

1. If the authors have adequately addressed your comments raised in a previous round of review and you feel that this manuscript is now acceptable for publication, you may indicate that here to bypass the “Comments to the Author” section, enter your conflict of interest statement in the “Confidential to Editor” section, and submit your "Accept" recommendation.

Reviewer #1: (No Response)

Reviewer #2: All comments have been addressed

2. Does this manuscript meet PLOS Global Public Health’s publication criteria? Is the manuscript technically sound, and do the data support the conclusions? The manuscript must describe methodologically and ethically rigorous research with conclusions that are appropriately drawn based on the data presented.

Reviewer #1: Yes

Reviewer #2: Yes

3. Has the statistical analysis been performed appropriately and rigorously?

Reviewer #1: Yes

Reviewer #2: Yes

4. Have the authors made all data underlying the findings in their manuscript fully available (please refer to the Data Availability Statement at the start of the manuscript PDF file)?

Reviewer #1: (No Response)

Reviewer #2: Yes

5. Is the manuscript presented in an intelligible fashion and written in standard English?

Reviewer #1: Yes

Reviewer #2: Yes

6. Review Comments to the Author

Reviewer #1: Major comments

1. This study actually compared the rates of two time points without using proper controls and this might be the major limitation of this study. If authors can show the results using negative controls to back up the causal interpretation of the study findings, it may increase the overall internal validity of the study.

2. Is there any other possible regulations, governmental policies, and societal changes which might affect the study findings other than implementation of regional trauma system? These contents should be reviewed and discussed in the manuscript.

3. Related to the response for comment no 1, authors should explain the time lag between master plan development (2012) and actual implementation of national trauma system in detail and explain the reasons for using only two time points in the manuscript.

Minor comments

1. The resolution of all figures should be improved.

2. Page 7 line 125 Hope authors change the ‘Database used for the study’ to ‘Data used for the study’ and move the contents related to panel review into this section.

3. Page 7 line 139, More explanation is needed for the WHO “Guidelines for trauma quality improvement programmes” and its relation to panels’ decision of preventability.

4. Page 8 line 162-163 More explanations are needed for ICISS and TRISS score model. (Definition, components, how they are calculated and their differences, meaning and etc). In addition, proper references should be placed.

5. Preventable trauma death rate from survey data for year 2015 and 2017 can be described in monthly plot. Parallel trends are expected and authors may check whether there are specific patterns among two years of monthly plot.

6. Page 11 line 233-235 This sentence should move to the discussion section.

7. Overall summary should be placed in the start of the discussion section.

Reviewer #2: The authors of the manuscript provide results and a thorough analysis of a country-wide evaluation of emergency medical services focused on trauma care in South Korea. The government is currently implementing a new, nationally regulated emergency trauma care system, which is a subject of periodic formal evaluation. The authors collect the results of the first two evaluations, both nationwide, and rigorously analyze the effectiveness of the trauma care system under development. Their analysis indicates that the new emergency care significantly improved the treatment of trauma patients, with a substantial decrease in the preventable death rate. The contributions of this manuscript are both in the methodology, data, and the evaluation of the improvement strategy South Korea is currently establishing. As such the manuscript can be of broad use to many professionals from organizing similar panels to choosing the right evaluation methodology. Finally, the improvement model can serve as a successful example to other countries, especially ones with an underdeveloped trauma care system. Therefore, I highly recommend this manuscript for publication.

Suggested minor revisions are:

1) On page 4, line 57 the authors state that 10% of people die from injury. I suggest rephrasing this sentence to stress out that it is 10% of all deaths are due to injury.

2) On page 4, lines 69-70 the authors should consider elaborating the statement on low-quality evidence with some more details, perhaps stressing out how the evidence in their study is more reliable.

3) Authors may consider mentioning some other, similar studies, to put their work in more context. However, I think the importance of this publication is clear enough even without a more extensive review.

4) The authors should improve the resolution of their figures. Figures 1 and 5 are too blurry to be read, and the remaining figures would benefit from a more sharp display (though they are readable).

7. PLOS authors have the option to publish the peer review history of their article (what does this mean?). If published, this will include your full peer review and any attached files.

**Do you want your identity to be public for this peer review?** For information about this choice, including consent withdrawal, please see our Privacy Policy.

Reviewer #1: No

Reviewer #2: No

---

## [Editor Report · Decision Letter 2]

22 Dec 2021

National trauma system establishment based on implementation of regional trauma centers improves outcomes of trauma care; A follow-up observational study in South Korea

PGPH-D-21-00240R2

Dear Dr. Kim,

We're pleased to inform you that your manuscript has been judged scientifically suitable for publication and will be formally accepted for publication once it meets all outstanding technical requirements.

Within one week, you'll receive an e-mail detailing the required amendments. When these have been addressed, you'll receive a formal acceptance letter and your manuscript will be scheduled for publication.

An invoice for payment will follow shortly after the formal acceptance. To ensure an efficient process, please log into Editorial Manager at https://www.editorialmanager.com/pgph/ click the 'Update My Information' link at the top of the page, and double check that your user information is up-to-date. If you have any billing related questions, please contact our Author Billing department directly at authorbilling@plos.org.

Kind regards,

Jagdish Khubchandani

Academic Editor

Additional Editor Comments (optional):

Thank you for all the revisions and clarifications